# Image Inpainting with Parallel Decoding Structure for Future Internet

**Peng Zhao [1], Bowei Chen [2], Xunli Fan [2], Haipeng Chen [3],\* and Yongxin Zhang [1],\***

[1] School of Information Technology, Luoyang Normal University, Luoyang 471022, China
[2] School of Information Science and Technology, Northwest University, Xi'an 710127, China
[3] School of Electrical Engineering, Northeast Electric Power University, Jilin 132012, China
\* Correspondence: haipeng0704@126.com (H.C.); zyx@lynu.edu.cn (Y.Z.)

**Abstract:** Image inpainting benefits much from the future Internet, but the memory and computational cost in encoding image features in deep learning methods poses great challenges to this field. In this paper, we propose a parallel decoding structure based on GANs for image inpainting, which comprises a single encoding network and a parallel decoding network. By adding a diet parallel extended-decoder path for semantic inpainting (Diet-PEPSI) unit to the encoder network, we can employ a new rate-adaptive dilated convolutional layer to share the weights to dynamically generate feature maps by the given dilation rate, which can effectively decrease the number of convolutional layer parameters. For the decoding network composed of rough paths and inpainting paths, we propose the use of an improved CAM for reconstruction in the decoder that results in a smooth transition at the border of defective areas. For the discriminator, we substitute the local discriminator with a region ensemble discriminator, which can attack the restraint of only the recovering square, like areas for traditional methods with the robust training of a new loss function. The experiments on CelebA and CelebA-HQ verify the significance of the proposed method regarding both resource overhead and recovery performance.

**Keywords:** image inpainting; generative adversarial networks (GANs); contextual attention module; discriminator; future Internet

## 1. Introduction

Image inpainting originated in the Renaissance era when damaged images were repaired through manual trimming to fill in the missing areas with adequate information. Today, this technique involves inferring unknown areas of an image with known information using methods such as structural, statistical, and semantic analysis. Due to its versatility and usefulness, image inpainting has become an increasingly popular research area in the field of computer vision. As the future Internet officially ushers in the second stage of its development, image inpainting will face a huge challenge. This technique has already been applied in different fields, such as medicine, the military, and video processing, among others [1–6]. Traditional methods for image inpainting often have some defects. However, with the rapid rise of deep learning in computer vision, the image processing methods based on this technology have significantly improved its effectiveness. For example, deep-learning-based image inpainting methods use network models such as convolutional neural networks (CNNs), generative adversarial networks (GANs), and recurrent neural networks (RNNs), and network modules such as attention mechanisms and residual networks [7–9]. Image inpainting can be classified into three types: traditional methods, CNN-based methods, and GAN-based methods.

The traditional image inpainting techniques can be categorized into two types: diffusion-based [10,11] and patch-based methods [12,13]. The diffusion-based method involves gradually diffusing the pixel information surrounding the damaged area in the image and creating new textures to fill the hole. This method is effective in restoring small

missing areas in the image. However, the reconstruction process is often influenced by the surrounding information, making it difficult to learn from distant information. Additionally, this method lacks a high-level semantic understanding of the image and may not successfully restore meaningful texture structures in the missing area. As the diffusion distance of pixel information around the hole increases, the larger the hole is, and the less effective the pixel information becomes in the center.

The patch-based method assumes that the image's missing area has equivalent content to the identified area. It seeks the most relevant matching patch in the visible area of the image before copying that information to fill the missing area at the pixel level. However, in most cases, the content of the damaged section of the image is quite different and unstructured from the local damage. Consequently, relevant patches may not be found in the image. To address this, researchers have proposed image inpainting between images. This approach mainly involves locating pictures similar in meaning to the target damaged image in an existing image database. The appropriate patch information is then selected and transplanted or borrowed [14]. This type of method provides better image inpainting of damaged images, especially when available image data are abundant within a specific domain. However, it requires a considerable amount of domain data acquisition and a best-match search, making it applicable only to a limited range of scenarios.

With the development of the field of CNN models, the impact of neural networks on inpainting has seen significant improvements [15]. Numerous CNN models have emerged since then, with the network structure and depth being frequently optimized and modified. One such model that has proven advantageous is the symmetrical U-Net structure, introduced in 2015. The left side is used for feature extraction, while the right side is used for upwards sampling. U-Net requires fewer samples and has a lower error rate, making it ideal for medical image processing and picture inpainting. Liu et al. [16] drew inspiration from image inpainting, presenting a load missing data recovery problem as a load image inpainting problem. To restore incomplete load images, they introduced a residual network (ResNet) and a convolutional block attention module (CBAM) to improve U-Net, enabling the efficient recovery of load missing data based on an actual industrial load dataset. Similarly, in 2021, Zeng et al. [17] sought to address limitations in deep generative models concerning inpainting output control and output diversity. Their novel free-form image inpainting framework uses a U-Net-like convolutional neural network to map an input to a coarse inpainting output. Pixel-wise matching based on nearest neighbors is applied to map the coarse output to multiple, high-quality outputs in a controllable manner. Their method offers multiple outputs with higher diversity, providing a promising alternative for future studies.

The emergence of GANs as a powerful technique for unsupervised learning in complex data distributions has been one of the most promising developments in recent years. This technique, proposed by Goodfellow et al. in 2014, allows the creation of models through adversarial processes that use noise to generate identical objects not present in the database [18]. GANs have proven particularly effective in generating images, and researchers are now exploring their potential to reproduce damaged areas in photos. However, the former structure contains two stages and involves a large number of parameters, which is a heavy burden to overcome. The advent of the one-stage network, especially the invention of Diet-PEPSI, alleviates the problem greatly. Thus, it is natural for us to employ this structure in our inpainting network to shrink the size of the model. On the other hand, there are works focused on improving the image generation results of GANs. In 2016, Pathak et al. proposed a network known as context encoder that combines the encoder and a GAN to perform unsupervised feature learning for image inpainting with large missing areas, resulting in more natural image restoration [19]. However, this approach is limited to filling square holes in the center of the image. In 2017, inspired by the context encoder, researchers developed a globally and locally consistent image inpainting method that addresses some of the context encoder's limitations, such as the ability to process only fixed low-resolution images, with the mask area located only in the center of the

image, and an inability to maintain local consistency with the surrounding area. In 2018, Yu et al. extended prior methods by incorporating a traditional coarse-to-fine network and a contextual attention module (CAM) [20]. The CAM enables the network to focus on filling in any missing portions, while the coarse network is used to concentrate on the image's edge information and the fine network is utilized for inpainting. While this method improves picture inpainting and enhances performance, GAN's inherent instability and high computational requirements remain significant challenges that need to be addressed. In summary, GANs continue to be a very promising technique for unsupervised learning in complex distributions, and researchers have developed several effective methods for picture inpainting. However, addressing GAN's limitations remains a major challenge for researchers, especially regarding stability and computational requirements.

The surge in internet transmission capacity and the widespread use of mobile cameras have led to a surge in demand for high-resolution images and videos [21–23]. However, traditional GAN-based image inpainting methods suffer from unstable training processes, insufficient diversity of generated samples and difficulty in terms of tuning the model parameters. Thus, they have great potential for improvement. Attention mechanisms have proven to be able to improve the performance and efficiency of deep neural networks. Researchers introduced the CAM to improve image inpainting quality with satisfactory results. However, it requires huge computational resources due to its two-stage process for feature encoding. Despite recent improvements to this drawback, it still faces significant challenges due to the high computational resources required, including convolution operations, network parameters, and poor output images. It still needs further research.

This paper aims to reduce the extensive consumption of computational resources resulting from feature encoding in the traditional coarse-to-fine network while improving inpainting performance. The primary contributions of this work include:

(1) The presentation of an image inpainting method based on a parallel decoding network that features a coarse path for preliminary inpainting results and an inpainting path for higher-quality inpainting results. This Diet-PEPSI structure can effectively decrease the number of parameters and lighten the network.

(2) The utilization of a truncated distance similarity metric in the CAM can enhance the performance of semantic feature representation. The overall and local semantic continuity are both significantly improved, with more natural and smooth results in the defective area.

(3) The replacement of the local discriminator with the region ensemble discriminator can enable the network to fit with more kinds of defected areas, as against the former rectangle area.

The rest of this paper is structured as follows. The basic concepts of generating adversarial networks and attention mechanisms are covered in Section 2. The proposed inpainting technique is introduced and described in Section 3. Section 4 includes the experimental setup, results, and discussion. Conclusions and future work directions are presented in Section 5.

## 2. Related Work

### 2.1. GAN

GANs consist of a generative network and a discriminant network. The generative network samples randomly from the noise distribution as the input, and its output should try to imitate the real samples in a given training set. The input of the discriminant network is the output or real sample of the generated network, whose goal is to separate the output of the generative network from the real sample as accurately as possible, while the generative network should deceive the discriminant network as much as possible. The two networks are against each other, constantly optimizing and adjusting the parameters, with the end goal of preventing the discriminant network from determining whether the output result of the discriminant network is accurate. The architecture of a GAN is shown in Figure 1.

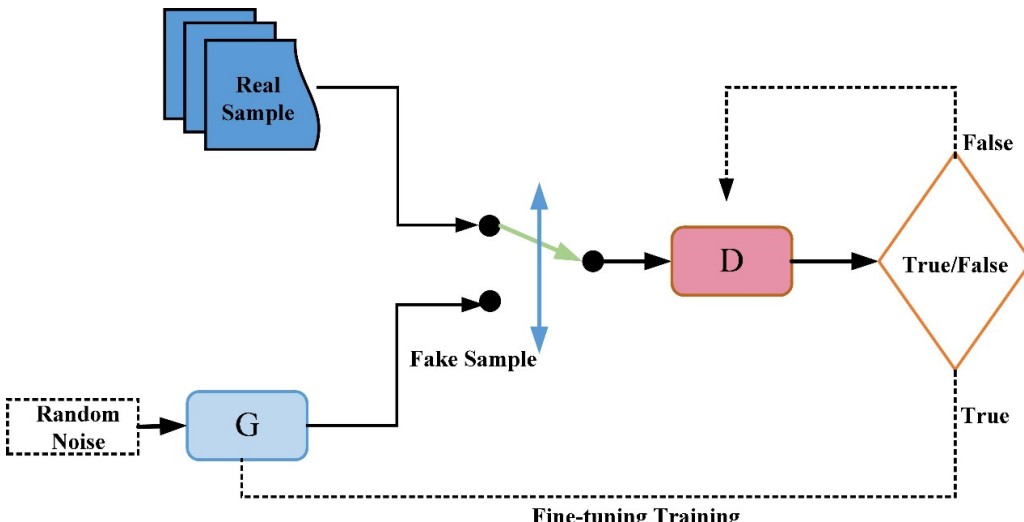

**Figure 1.** Architecture of GAN.

The training process of a GAN can be expressed by the following formula:

$$\max_{D} \min_{G} GAN(D,G) = E_{X \sim P_{data}(x)}[\log D(x)] + E_{Z \sim P_z(z)}[\log(1 - D(x))], \tag{1}$$

The generator, $G$, captures the distribution of the sample data, generates a sample similar to the real training data with noise $z$ following a specific distribution (uniform distribution, Gaussian distribution, etc.), and pursues the effect that the more similar to the real sample, the better.

The discriminator, $D$, is a binary classifier that estimates the probability of a sample being a real image. If the sample is from the real training data, $D$ will output a value to determine the effect of the generating images. The higher the value, the more real the image.

Compared with all other models, GANs can produce clearer and more real samples and are applied to some scenarios, such as super-resolution reconstruction, image editing, data generation, malicious attack detection, and avoiding the design of complex loss functions, and two networks will spontaneously fight against each other. However, GANs also have some aspects that need to be improved. Since GANs originate from the idea of a zero-sum game in game theory, the gradients of the generator and discriminator cancel each other, which leads to the difficulty of network convergence. In addition, a GAN itself has poor stability and is difficult to train. The context attention module can make a GAN network abandon the current irrelevant information and pay more attention to the target information, which can effectively reduce the complexity of training and improve the effect of inpainting.

*2.2. Attention Mechanism*

Attention mechanisms have evolved from the study of human vision. In cognitive science, the mechanisms by which people selectively focus on a portion of all information and ignore the rest are often referred to as attention mechanisms. One of the functions of attention mechanisms is to determine the part to focus on based on the query and then allocate more resources to this part. This resource allocation method can help the network process inputs more efficiently and improve the performance and efficiency of the network, especially when dealing with large-scale datasets and complex models [24,25]. Most of the time, deep neural networks need to focus on a certain part of the data, so the attention mechanism is introduced into deep learning. Contextual attention can help the network scan the overall image area and lock and pay more attention to the designated key areas. In essence, contextual attention is similar to human attention mechanisms. Its main purpose is to select the information that is more relevant to the current task objective from a large

amount of information, set dynamic weighting parameters, strengthen the context-related key information, weaken the irrelevant information, and greatly improve the efficiency of deep learning algorithms.

Context attention can judge where to obtain useful information and determine which feature information to use to patch the image. It is realized through convolution and matches the generated patch with the context patch. Softmax is used to balance the patch effect of deconvolution reconstruction using the context attention mechanism. The useful key information in the network is strengthened, while the relatively useless information is weakened so as to improve the learning efficiency in the network, which has been widely used in image inpainting algorithms. Yan et al. [26] proposed a shift network driven by shift and loss bootstrap operations, which obtained the relationship between the context region in the encoder layer and the corresponding missing region in the decoder layer. Song et al. [27] added a patch-swap module into the network, extracted feature maps through a VGG network, spread the texture details with the highest frequency from the boundary to the hole, and replaced the feature blocks corresponding to the missing areas of each feature map with the most similar feature blocks in the context area. The method in [23] adds the spatial propagation layer and uses the attention mechanism to increase the global consistency of the image but fails to repair the correlation between the missing areas well.

As is known, although GANs are powerful generative models that generate high-quality samples, their training process is unstable, the diversity of the generated samples is insufficient, and the difficulty of tuning the model parameters still need further research. The GAN-based image inpainting method has great potential for improvement. Attention mechanisms have improved the performance and efficiency of deep neural networks. Researchers introduced the CAM to improve image inpainting quality with satisfactory results. However, it requires huge computational resources due to its two-stage process for feature encoding. In order to develop more efficient and robust attention mechanisms to cope with this problem, we propose a parallel decoding network that features a coarse path for a preliminary effect and an inpainting path for a higher-quality effect. A truncation distance is applied to determine similarity scores in the improved CAM to enhance the performance of semantic feature representation. RED is used to replace the local discriminator in guiding the generator to obtain better results. The experiment indicates that the proposed method exhibits favorable performance in improving inpainted image quality and significantly reducing the computational time. The inpainted image shows enhanced semantic continuity and improved facial features, with smoother lines and visible muscle texture, resulting in higher consistency with the source image.

## 3. Methodology

### 3.1. Model Frame

#### 3.1.1. Network Model

Parallel extended-decoder path for semantic inpainting (PEPSI) is a parallel network [28] that uses a single-stage encoder–decoder structure. It utilizes a single encoding network to extract features and a single decoding network to generate a high-quality inpainted result. To aggregate contextual information, the network employs a series of dilated convolutional layers that contain a large number of network parameters. Although pruning the channels of these layers is an intuitive way to reduce the hardware cost, it often results in inferior performance in practical applications. The generative network consists of a shared encoding network and a parallel decoding network. The encoding network has a Diet-PEPSI unit [29], which helps to reduce the parameters in the network. To address this issue of PEPSI, our network utilizes rate-adaptive dilated convolutional layers, which share weights while producing dynamic feature maps according to the given dilation rates. The rate-adaptive dilated convolutional layers modify the shared weights by applying different scaling and shifting operations based on the given dilation rates. Encoding networks use a co-learning approach with coarse path and inpainting path parallel decoding networks. A CAM is

added to the inpainting path to improve the global consistency of the final image [30]. Since the two paths are parallel, the number of convolution operations is reduced. The discriminant network uses the region ensemble discriminator (RED) to process any missing region. RED can detect a target object appearing anywhere in the image by individually processing multiple feature regions. It utilizes individual regressors for each pixel, enabling its function as both a global and local discriminator at the same time. The architecture of the proposed network model is shown in Figure 2. Improvement is achieved in both the coarse and inpainting paths through weight-sharing. The L1 reconstruction loss is exclusively employed in the training of the coarse path, while both the L1 and adversarial losses are utilized in the training of the inpainting path.

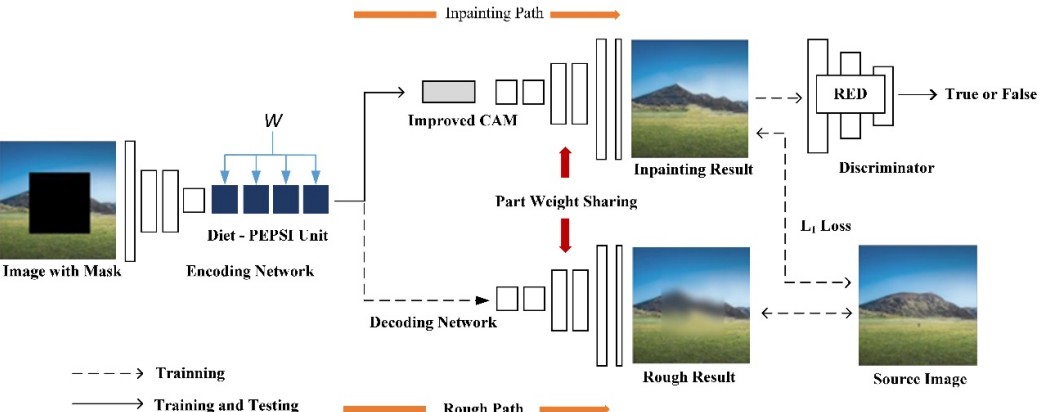

**Figure 2.** Architecture of the proposed network model.

### 3.1.2. Encoding Network

An encoding network is used to extract the features from the input images with missing areas and complete missing features. A Diet-PEPSI unit is added to the encoding network, which consists of a residual attentional spatial pyramid pooling convolutional layer (RASPP) of $3 \times 3$ and a standard convolutional layer of $1 \times 1$, as shown in Figure 3. We substituted the multiple expanded convolutional layers with a Diet-PEPSI unit. The Diet-PEPSI unit features convolutional layers that are adaptable to the pace of change, and they share weights, while the $1 \times 1$ standard convolutional layers do not. The whole feature encoding network consists of a series of $3 \times 3$ convolutional layers. In the first convolutional layer, a $5 \times 5$ convolutional kernel is used to make full use of the potential information in the input image, as shown in Table 1. In addition, we use the adaptive-rate expansion convolution layer in the last four convolution layers for scaling and shifting according to the given expansion rate and changing the shared weight.

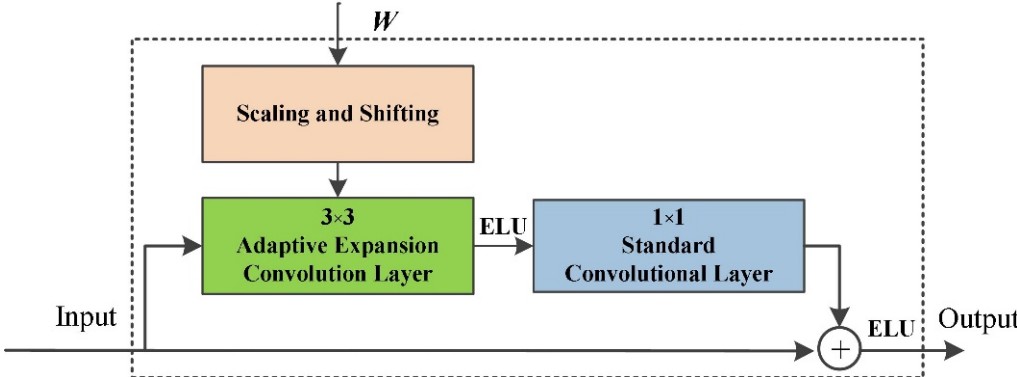

**Figure 3.** Architecture of Diet-PEPSI unit.

**Table 1.** Detailed architecture of shared encoding network.

| Type | Kernel | Dilation | Stride | Outputs |
|---|---|---|---|---|
| Convolution | $5 \times 5$ | 1 | $1 \times 1$ | 32 |
| Convolution | $3 \times 3$ | 1 | $2 \times 2$ | 64 |
| Convolution | $3 \times 3$ | 1 | $1 \times 1$ | 64 |
| Convolution | $3 \times 3$ | 1 | $2 \times 2$ | 128 |
| Convolution | $3 \times 3$ | 1 | $1 \times 1$ | 128 |
| Convolution | $3 \times 3$ | 1 | $2 \times 2$ | 256 |
| Dilated Convolution | $3 \times 3$ | 2 | $1 \times 1$ | 256 |
| Dilated Convolution | $3 \times 3$ | 4 | $1 \times 1$ | 256 |
| Dilated Convolution | $3 \times 3$ | 8 | $1 \times 1$ | 256 |
| Dilated Convolution | $3 \times 3$ | 16 | $1 \times 1$ | 256 |

### 3.1.3. Decoding Network

The parallel decoding network consists of a coarse path and an inpainting path. The coarse path produces a roughly complete result from the encoded feature map. On the other hand, taking the encoded feature as the input, the inpainting path first reconstructs the feature map using the CAM. Then, the reconstructed feature maps are decoded to generate higher-quality inpainting results. By sharing the weight parameters of the two paths, we attempt to normalize the repair path of the decoded network. In addition, two different paths use the same encoded feature map as their input, so they force a single encoder to generate valuable features for two different image generation tasks. During the implementation of the parallel decoding network, all convolutional layers are filled. In addition, the activation function ELU is adopted for all layers except the last layer, and the image is normalized to $[-1, 1]$, as shown in Table 2. The output layer comprises a convolutional layer that limits the value to the range of $[-1, 1]$.

**Table 2.** Detailed architecture of parallel decoding network.

| Type | Kernel | Dilation | Stride | Outputs |
|---|---|---|---|---|
| Convolution $\times 2$ | $3 \times 3$ | 1 | $1 \times 1$ | 128 |
| Upsample ($\times 2\uparrow$) | - | - | - | - |
| Convolution $\times 2$ | $3 \times 3$ | 1 | $1 \times 1$ | 64 |
| Upsample ($\times 2\uparrow$) | - | - | - | - |
| Convolution $\times 2$ | $3 \times 3$ | 1 | $1 \times 1$ | 32 |
| Upsample ($\times 2\uparrow$) | - | - | - | - |
| Convolution $\times 2$ | $3 \times 3$ | 1 | $1 \times 1$ | 16 |
| Convolution (output) | $3 \times 3$ | 1 | $1 \times 1$ | 3 |

### *3.2. Network Improvement*

#### 3.2.1. Diet-PEPSI Unit

In traditional networks, a series of dilated convolution layers with different dilation rates are often added to the encoding network to aggregate context information and extract features with large receptive fields. This requires a lot of network parameters. Intuitively we can reduce hardware costs by trimming the channels in these layers, but in practice, it usually produces poor results. To solve this problem, a new rate-adaptive dilated convolution layer is applied, which dynamically generates feature maps by using shared weights according to given dilation rates. Since the rate-adaptive layer shares the weight in each layer, the number of network parameters is significantly reduced compared with multiple standard extended convolutional layers.

In general, the weight of the convolution layer is considered to be a four-dimensional tensor $W \in R^{k \times k \times Cin \times Cout}$, where k is the kernel size, $C_{in}$ and $C_{out}$ are the number of input and output channels, respectively. The weights in each convolutional layer can be regarded as $C_{out}$ filters with $C_{in}$ channels. In order to generate different feature maps according to the given dilation rate, we define the scaling factor $\gamma_d \in R^{1 \times 1 \times Cin \times Cout}$ and the migration scale

$\beta_d \in \mathrm{R}^{1\times1\times\mathrm{Cin}\times\mathrm{Cout}}$ to adjust the parameter *W*, where d represents the dilation rate, $\gamma_d$ and $\beta_d$ is carried out according to the given dilation rate. The rate-adaptive scaling and shifting operations mentioned above are shown in Figure 4. The scaling and shifting operations include tensor broadcasting and $\beta_d$ as well as $\gamma_d$ take on varying values depending on the given rate. The adjustment formula of W is as follows:

$$W_d = \gamma_d \times W + \beta_d \tag{2}$$

where $W_d \in \mathrm{R}^{\mathrm{k}\times\mathrm{k}\times\mathrm{Cin}\times\mathrm{Cout}}$ represents the weight after rate-adaptive modification.

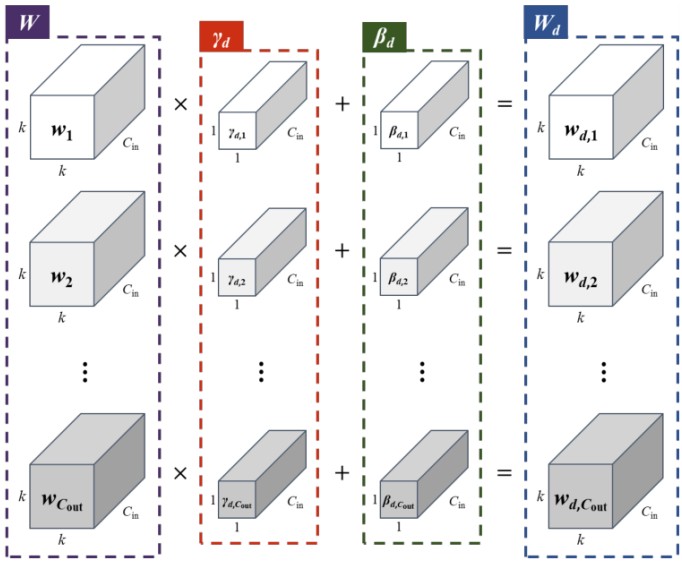

**Figure 4.** Rate-adaptive scaling and shifting operations.

Diet-PEPSI has the same receptive field size covered by the standard dilated convolutional layer, which requires $3 \times 3 \times \mathrm{C_{in}} \times \mathrm{C_{out}} \times \mathrm{n}$ parameters. The Diet-PEPSI unit requires $(9 + 3\mathrm{n}) \times \mathrm{C_{in}} \times \mathrm{C_{out}}$ parameters, where n represents the number of Diet-PEPSI units and expanded convolutional layers. When n is greater than 1, i.e., when multiple dilated convolution layers are required, it can be calculated that the network parameters brought by Diet-PEPSI are significantly reduced.

### 3.2.2. Improved CAM

The CAM divides the input feature map into foreground patches and background patches and measures the similarity score by applying cosine similarity. Finally, Softmax [31] was used to normalize the cosine similarity. The illustration of the CAM is shown in Figure 5. The conventional CAM reconstructs the foreground patches by measuring the cosine similarities with the background patches. In contrast, the modified CAM uses the Euclidean distance to compute similarity scores. However, the normalization of the feature block vector in Formula (3) will distort the semantic feature representation and lead to distortion.

$$s_{(x,y),(x',y')} = \left\langle \frac{f_{x,y}}{\|f_{x,y}\|}, \frac{b_{x',y'}}{\|b_{x',y'}\|} \right\rangle$$
$$s^*{}_{(x,y),(x',y')} = soft\mathrm{max}(\lambda s(x,y),(x',y')) \tag{3}$$

where $f_{x,y}$ are the coordinates of the foreground patching, $b_{x',y'}$ are the coordinates of the background patch, and $s(x,y),(x',y')$ are the similarity scores. $\lambda$ is the hyperparameter of the scaled softmax, $s^*(x,y),(x',y')$ is the weighted sum of the weighted background blocks.

To address the above problems, we use an improved CAM that uses Euclidean distance to directly measure distance similarity scores because Euclidean distance takes into account not only the angle between two feature block vectors but also their magnitude. However, directly using Euclidean distance is not conducive to Softmax normalization. Truncation

distance similarity is a metric used to measure the quality of a generative model's output in image synthesis. It measures the dissimilarity between the distribution of generated samples and the distribution of real samples by comparing their truncated principal component analysis (PCA) representations. The truncation distance is computed by taking the average distance between the truncated PCA representations of generated and real samples. A smaller truncation distance indicates the proximity of the generated samples to the real samples in terms of their distribution, and as a result, truncation distance similarity is used as a metric for measuring the quality of a generative model's output. Truncation distance similarity is as follows:

$$
\begin{aligned}
\widetilde{d}_{(x,y),(x',y')} &= \tan h\big(-\big(\tfrac{d_{(x,y),(x',y')}-m(d_{(x,y),(x',y')})}{\sigma(d_{(x,y),(x',y')})}\big)\big) \\
d_{(x,y),(x',y')} &= \|f_{x,y} - b_{x',y'}\|
\end{aligned}
\tag{4}
$$

$f_{x,y}$ are the coordinates of the foreground patching, $b_{x',y'}$ are the coordinates of the background patch, $d_{(x,y),(x,y)}$ are the similarity scores, and $\widetilde{d}_{(x,y),(x',y')}$ are truncation distances. By using Euclidean distance instead of cosine similarity in the intercept design, not only the vector angle is taken into account, but also the magnitude is considered. Images reconstructed by applying the truncated distance similarity can collect more similar patches than the cosine similarity.

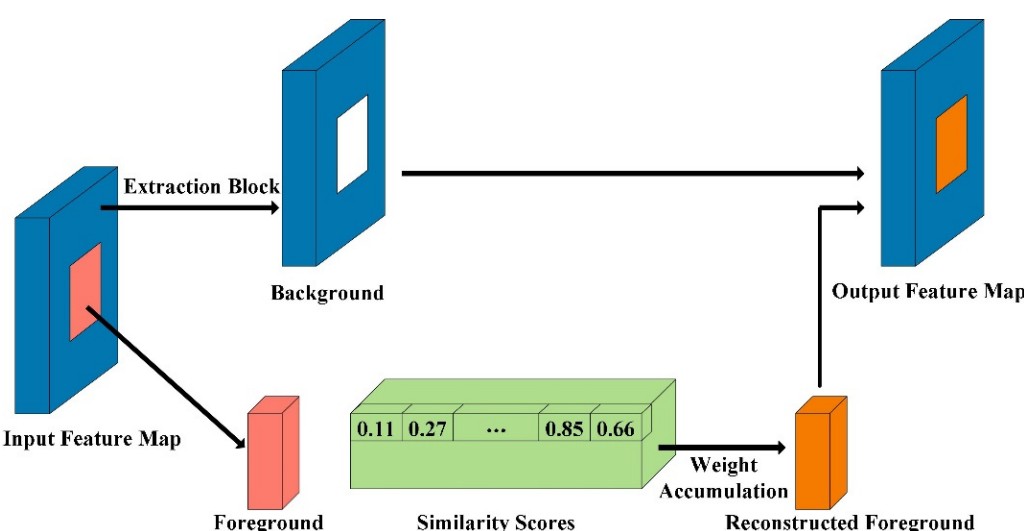

**Figure 5.** Illustration of CAM.

### 3.2.3. RED

Traditional global and local discriminators consider not only the coherence of the entire image but also the local texture of the missing area. However, the local discriminator can only deal with the missing area of the size of a square. To solve this problem, the global and local discriminators can be unified into the regional ensemble discriminator. RED consists of six convolution kernel with a size of 5 × 5 and a stride of two and a final separate fully connected layer [32], as shown in Table 3. RED divides the last layer of features into pixel-level blocks, with each pixel using a separate fully connected layer that separately discriminates each block. The illustration of RED is shown in Figure 6. In the final layer, each pixel uses a fully connected layer with unique weights to classify any defected regions that may be present in an image, regardless of their size or location. Since RED tries to classify each feature block with different receptive fields separately, it judges different areas of the image separately. In contrast to local discriminators, RED can handle a variety of missing areas that may appear anywhere in an image of any size. After each convolution layer, except for the last, there is a Leaky-ReLU as the activation function that normalizes each layer [33].

**Table 3.** Detailed structure of regional integration discriminator model.

| Type | Kernel | Stride | Outputs |
| --- | --- | --- | --- |
| Convolution | $5 \times 5$ | $2 \times 2$ | 64 |
| Convolution | $5 \times 5$ | $2 \times 2$ | 128 |
| Convolution | $5 \times 5$ | $2 \times 2$ | 256 |
| Convolution | $5 \times 5$ | $2 \times 2$ | 256 |
| Convolution | $5 \times 5$ | $2 \times 2$ | 256 |
| Convolution | $5 \times 5$ | $2 \times 2$ | 512 |
| FC | $1 \times 1$ | $1 \times 1$ | 1 |

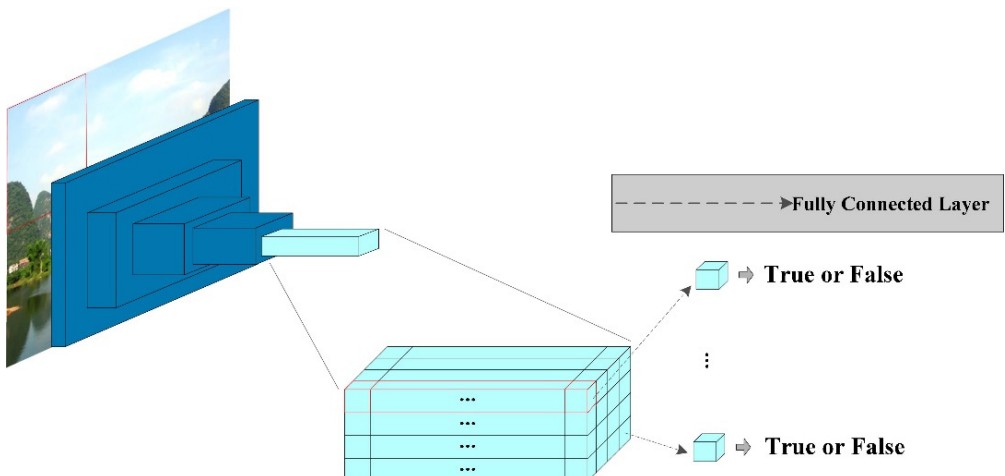

**Figure 6.** Illustration of RED.

*3.3. Design of Loss Function*

In traditional GANs, the generator is prone to gradient disappearance [34]. In order to solve this problem, inspired by the literature [23], the network model in this study uses GAN loss functions of the generator and discriminator. To train PEPSI, we jointly optimize two different paths in the decoding phase: the inpainting path and the coarse path.

For inpainting paths, L1 losses and GAN losses are used. The formula is as follows:

$$L_G = -E_{x \sim p_{x_i}}[D(x)] \tag{5}$$

$$L_D = E_{x \sim P_y}[\min(0, -1 + D(x))] - E_{P_{x_i}}[\min(0, -1 - D(x))] \tag{6}$$

where $P_{xi}$ and $P_y$ represent the data distribution of the inpainting results and input images. Since the goal of image restoration is not only to fill the missing part but also to obtain a natural and smooth result, we use the L1 norm to add a strong constraint as a penalty term in Formula (7), as shown below:

$$L_G = \frac{\lambda_i}{N} \sum_{n=1}^{N} \|X_i^{(n)} - Y^{(n)}\|_1 - \lambda_{adv} E_{P_{x_i}}[D(x)] \tag{7}$$

where $X_i^{(n)}$ and $Y^{(n)}$ are, respectively, generated through the inpainting path of the image of the nth image pair and the original input image in its corresponding batch size, $N$ is the batch size of the image logarithm, and $\lambda_i$ and $\lambda_{adv}$ are the super parameters to balance between the two loss items.

For coarse paths, their role is to correctly complete the missing features for CAM, using L1 losses:

$$L_c = \frac{1}{N} \sum_{n=1}^{N} \|X_c^{(n)} - Y^{(n)}\|_1 \tag{8}$$

where $Xc^{(n)}$ and $Y^{(n)}$ are the nth image pair of the image generated through the coarse path and their corresponding small-batch original input images, respectively.

Finally, we define the total loss function of the PEPSI generation network as follows:

$$L_{total} = L_G + \lambda_c (1 - \frac{k}{k_{\max}}) L_c \qquad (9)$$

where $\lambda_c$ is the weight parameter and $k$ and $k_{\max}$ represent the iteration and maximum iteration periods of the learning process, respectively.

## 4. Experiments

### 4.1. Experimental Settings

The dataset used in this paper is CelebA Face [35], which contains more than 200,000 photos of faces with feature annotations on each. A portion of them, 1000 training sets and 100 test sets were chosen for the experiment. CelebA-HQ Face [36] is a high-resolution version of the CelebA Face dataset. It includes over 30,000 high-quality facial images, each with a resolution of $1024 \times 1024$ and the same 40 attribute annotations as the original CelebA Face dataset. Compared to CelebA Face, CelebA-HQ Face has higher image quality and more detailed features and is, therefore, more challenging. It is also more suitable for developing high-fidelity face generation algorithms and more fine-grained facial attribute analysis algorithms. To construct our training and test sets, we performed a random sampling of 28,000 images from CelebA Face and CelebA-HQ Face for the training set and 8000 images for the test set from the dataset. Figure 7 depicts a portion of the picture samples.

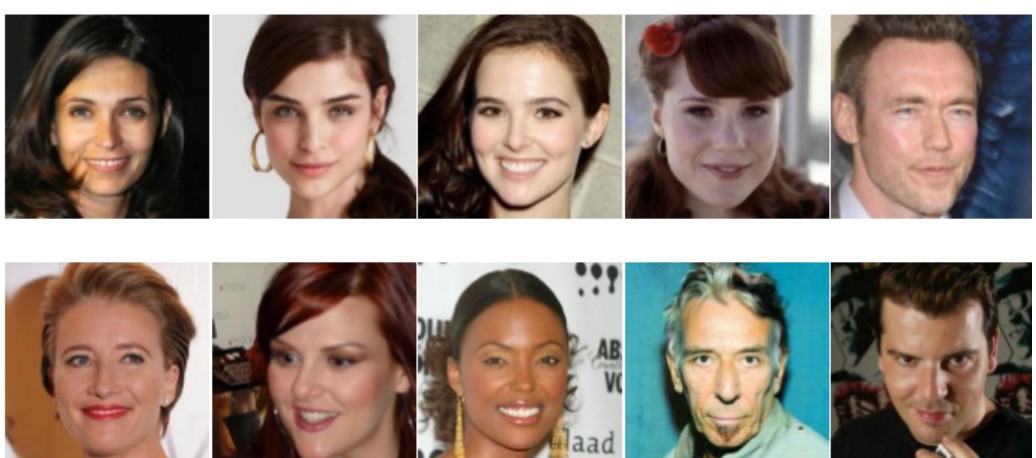

**Figure 7.** The part of the sample from CelebA and CelebA-HQ Face dataset.

We trained our models' 100,000 epochs with a batch size of 8, with learning rates of 0.0004 and 0.0001 for the discriminator and generator, respectively. The learning rate is reduced to 1/10 after 0.9 million iterations. The maximum number of iterations is 10,000. The mask size is $128 \times 128$. The image input resolution is $256 \times 256$. The optimize_ D learning rate is 0.0004. The optimize_ G learning rate is: 0.0001.

The operating system, Windows 10, a desktop PC with 32GB of internal memory, a Core i5-9600K CPU, and a GeForce GTX 1660 SUPER GPU are all used in this study. Tensorflow 1.15.0 was chosen as the framework for the experiment, together with Cuda 10.0, python 3.6.0, and Cudnn 7.6.1.

To showcase the significance of our proposed method, we conducted a comparison of their qualitative and quantitative operation speeds and the number of network parameters against the conventional generative methods, which include context encoders (CE) [22], a generator with contextual attention (GCA) [23], and PEPSI [28].

### 4.2. Evaluation Metrics

Since the subjective evaluation information is greatly affected by individuals, it is more objective and unified to use quantitative metrics to evaluate the image inpainting results [37–39]. The following are mainly used to evaluate the results of image restoration, mean square error (MSE) [40], peak signal-to-noise ratio (PSNR) [41], and structural similarity index (SSIM) [42].

The formula for PSNR is as follows:

$$
\begin{aligned}
MSE &= \frac{1}{H \times W} \sum_{i=1}^{H} \sum_{j=1}^{W} [X(i,j) - Y(i,j)]^2 \\
PSNR &= 10 \times \log_{10}\left(\frac{MAX_1^2}{MSE}\right) = 20 \times \log_{10}\left(\frac{MAX_1}{\sqrt{MSE}}\right)
\end{aligned}
\tag{10}
$$

where $H$ and $W$, respectively, represent the width and height of the image, $(i,j)$ represents each pixel of the image, $X(i,j)$ represents the pixel information value after inpainting the missing area of the image, $Y(i,j)$ represents the pixel information value of the missing area, and $MAX_1$ represents the maximum value of the color of the image point. The larger the $PSNR$ is, the less the distortion of the repaired image and the richer the diversity of the pixel information.

The calculation formula of SSIM is as follows:

$$
\begin{aligned}
l(x,y) &= \frac{2\mu_x \mu_y + C_1}{\mu_x^2 + \mu_y^2 + C_1} \\
c(x,y) &= \frac{2\sigma_x \sigma_y + C_2}{\sigma_x^2 + \sigma_y^2 + C_2} \\
s(x,y) &= \frac{\sigma_{xy} + C_3}{\sigma_x \sigma_y + C_3} \\
SSIM(x,y) &= l(x,y) \cdot c(x,y) \cdot s(x,y)
\end{aligned}
\tag{11}
$$

where $\mu_x$ represents the mean of sample $X$, $\mu_y$ represents the mean of sample $Y$, $\sigma_x$ represents the variance of sample $X$, $\sigma_y$ represents the variance of sample $Y$, $\sigma_{xy}$ represents the covariance of sample $X$ and sample $Y$, and $C_1$, $C_2$, and $C_3$ are all constants. The range of $SSIM$ is [0, 1]. The larger the $SSIM$ is, the more similar the structure of the two images is. In other words, the higher the quality of the inpainting image, the smaller the distortion.

### 4.3. Experimental Results and Analysis

#### 4.3.1. Diet-PEPSI Unit Validation

To compare the impacts before and after adding the Diet-PEPSI unit, we utilized the same generator and discriminator, the same loss function as the optimization goal, all of the parameters were set to the same value, and the environment was also the same. The image inpainting performance comparison with/without the Diet-PEPSI unit is shown in Figure 8.

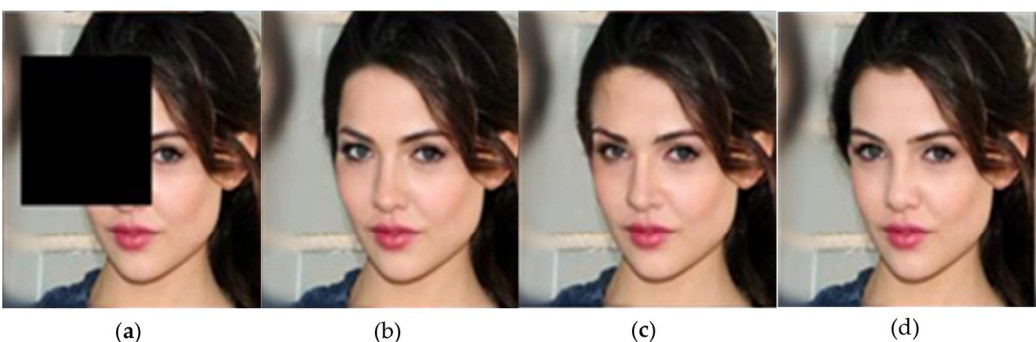

(a)          (b)          (c)          (d)

**Figure 8.** Image inpainting performance comparison with/without Diet-PEPSI unit. (**a**) Image with Mask; (**b**) Image with Mask; (**c**) Without Diet-PEPSI; (**d**) Source image.

As can be seen from the comparison effect, there is little difference in the effect of the model using Diet-PEPSI instead of dilated convolution. Because the Diet-PEPSI consists of an adaptive dilated convolutional layer and a standard convolutional layer, the adaptive dilated convolutional layer shares weights between each layer, greatly reducing the parameters in the network. This also means that using Diet-PEPSI can reduce memory usage without compromising the inpainting effect, reducing the hardware configuration requirements of the model.

### 4.3.2. Improved CAM Validation

To compare the impacts before and after the improvement of the CAM module, we utilized the same generator and discriminator, the same loss function as the optimization goal, all parameters were set to the same value, and the environment was also the same. The image inpainting performance comparison of CAM improvement or not is shown in Figure 9.

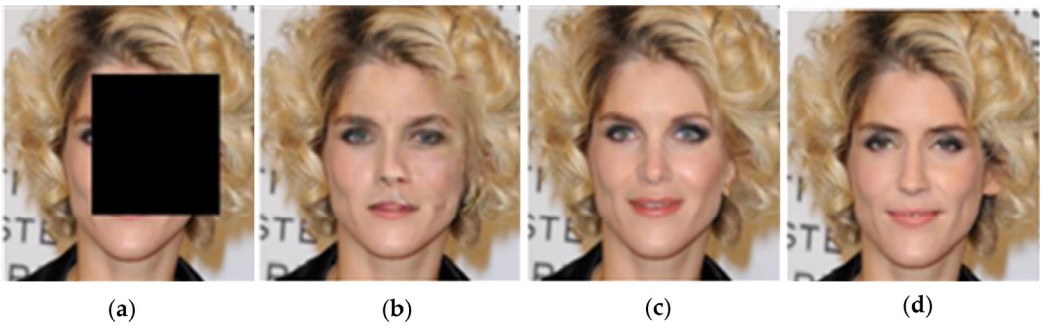

|  (a)  |  (b)  |  (c)  |  (d)  |

**Figure 9.** Image inpainting comparison of using CAM improvement or not. (**a**) Input image of the network; (**b**) Result of CAM; (**c**) Result of improved CAM; (**d**) Ground Truth.

It can be seen from the comparison effect that after improving the CAM, the inpainting effect is more similar to the real image. Because the improved CAM takes the size of the feature block into account in addition to the vector angle of the feature block, the image's overall semantic continuity and local semantic continuity have both significantly improved, and the smoother facial lines and muscle texture are clearly visible. Additionally, it indicates that the improved CAM is better suited to handle the relationship between the background and the defect area.

### 4.3.3. RED Validation

To compare the impacts of using the local discriminator and using the RED, we utilized the same generator and discriminator, the same loss function as the optimization goal, all of the parameters were set to the same value, and the environment was also the same. The inpainting results of using a local discriminator or RED are shown in Figure 10.

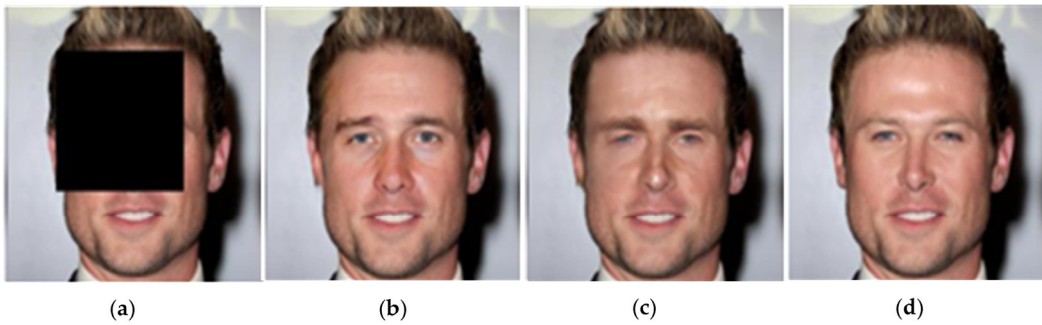

|  (a)  |  (b)  |  (c)  |  (d)  |

**Figure 10.** Inpainting results of using local discriminator or RED. (**a**) Input image of the network; (**b**) Result of using local discriminator; (**c**) Result of using RED; (**d**) Ground truth.

It can be seen from the comparison effect that the inpainting image is more consistent with the real image after RED replaces the local discriminator. The use of a local discriminator has some flaws in the details of image inpainting, relying too much on the missing areas of the image and ignoring the overall continuity of the image. On the other hand, RED judges each feature block in different receptive fields to improve image quality.

### 4.3.4. Qualitative Assessments

In the experiment, we, respectively, investigated the use of a square mask and a free mask for the input image. Figure 11 depicts the inpainting results of different methods with the square mask for the input image. Figure 12 depicts the inpainting results of different methods with the free mask for the input image.

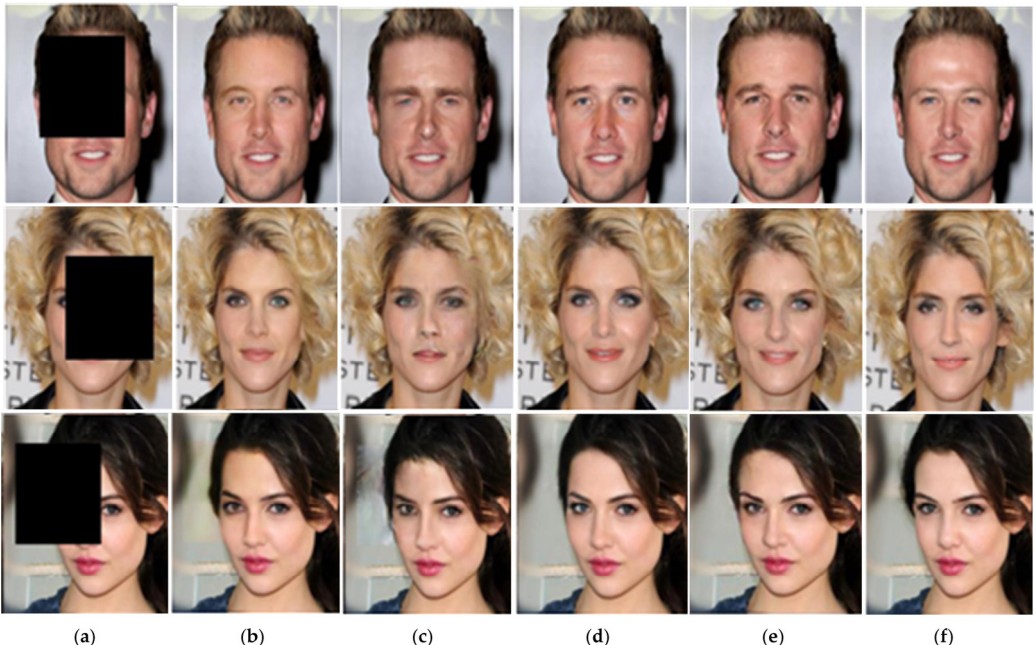

|     |     |     |     |     |     |
| --- | --- | --- | --- | --- | --- |
| (**a**) | (**b**) | (**c**) | (**d**) | (**e**) | (**f**) |

**Figure 11.** Comparison of different methods on the square masked images. (**a**) Input images of the network; (**b**) Results of CE; (**c**) Results of GCA; (**d**) Results of PEPSI; (**e**) Results of Ours; (**f**) Ground truth.

It is easy to see that these compared methods all can effectively inpaint the image with a square mask in Figure 11. No residual of the square mask appeared on the inpainted face images. However, blur and distortion appear on the inpainted face images of GCA, such as the blurred eyes and the distorted nose and lips of the woman in the middle row and the dead color of the left face of the woman in the bottom row. The CE method cannot well inpaint the image with a free mask, as seen in Figure 12. The residual of the free mask appears on the inpainted face image of CE, such as the forehead of the two men and the eyes and hair of the woman. The visual quality of PEPSI and the proposed method are similar to the source images and are all good. The detail features such as the edge, structure, textures, and colors of the inpainted face images obtained by the proposed method are better than that of PEPSI. The proposed method's single-stage network structure can overcome the limitations of the two-stage coarse-to-fine network by utilizing a joint learning scheme. It can drive the encoding network to properly produce missing features for the CAM by using the coarse path. The RED used in the proposed method is inspired by the region ensemble network, which is capable of classifying objects in any region of the image. This improves the performance of the generator on free-form masks and effectively drives the generator to produce visually pleasing inpainting results. The proposed method can obtain better visual quality inpainted images than the other three compared methods.

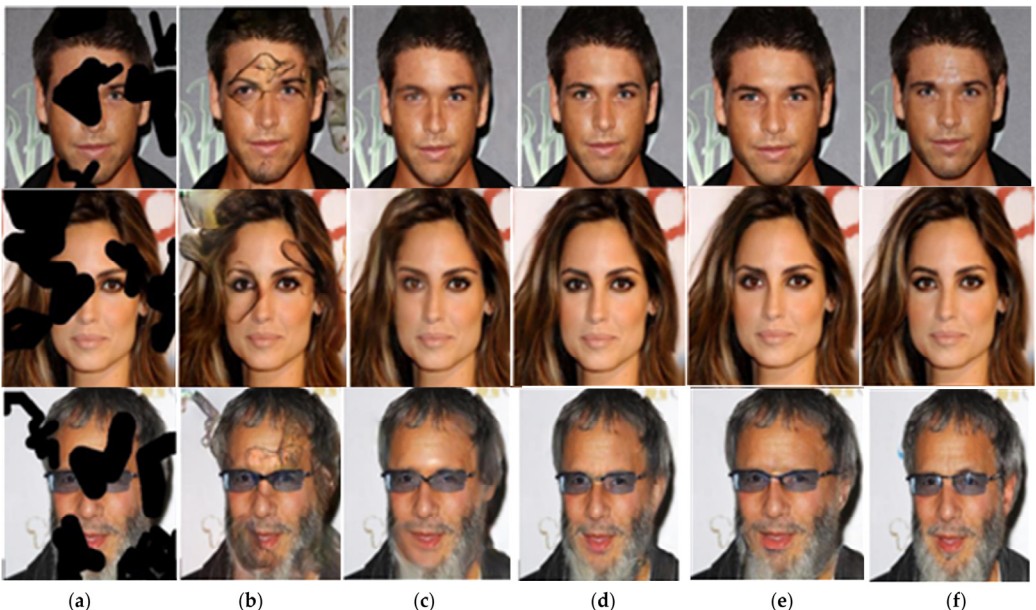

|     |     |     |     |     |     |
| --- | --- | --- | --- | --- | --- |
| (**a**) | (**b**) | (**c**) | (**d**) | (**e**) | (**f**) |

**Figure 12.** Comparison of different methods on the free-form masked images. (**a**) Input images of the network; (**b**) Results of CE; (**c**) Results of GCA; (**d**) Results of PEPSI; (**e**) Results of Ours; (**f**) Ground truth.

4.3.5. Quantitative Assessments

Two quality metrics mentioned above are performed on the inpainted images obtained by different comparison methods. The network parameter quantity (PQ) and running time (Time) are used to measure the efficiency of the comparison methods. The measured results of PSNR, SSIM, Time, and PQ are listed in Tables 4 and 5. Table 4 shows the measured results of the comparison methods for the square masked images. Table 5 shows the measured results of the comparison methods for the free-form masked images. It can be seen that, in Table 4, our model obtains the highest score in PSNR and SSIM. Compared with the CE model, our model's PSNR increases by 3.5 dB, SSIM increases by 0.006%, time is shortened by 10.6 ms, and the number of network parameters is reduced by 3.3 M. Compared with the GCA, PSNR, SSIM, time and parameter number of PEPSI increase by 0.6 dB, 0.005%, 1.0 ms and 0.4 M, respectively, with little overall change. Compared with PESPI, the PSNR of our model is increased by 0.4 dB, SSIM by 0.002%, time is shortened by 0.6 ms, and the PQ is reduced by 1.4 M. Except for the obvious reduction in network parameters, the other evaluation indexes have no significant changes. It can be seen in Table 5 that our model obtains the highest score in PSNR and SSIM. Compared with the CE, the PSNR of our model increases by 7.9 dB. The SSIM is increased by 0.029%, the time is shortened by 10.6 ms, and the number of network parameters is reduced by 3.3 M. Compared with GCA, the PSNR of PEPSI is increased by 4.4 dB, SSIM by 0.026%, time by 1.7 ms, parameter number by 0.4 M, and PSNR is greatly improved, and other indicators have little change. Compared with the PEPSI, the PSNR of our model is increased by 0.2 dB, SSIM by 0.003%, time is shortened by 0.8 ms, and the number of network parameters is reduced by 1.4 M. Except for the obvious reduction in network parameters, the other evaluation indexes have no significant change. In order to ease comparison, bar charts of the metric values in Tables 4 and 5 are shown in Figure 13. It is easy to see that the performance of the proposed method is superior to that of the other compared methods.

**Table 4.** Performance of the different inpainting methods for square masked images.

| Net Model | PSNR/dB | SSIM/% | Time/ms | PQ/M |
|-----------|---------|--------|---------|------|
| CE | 23.7 | 0.895 | 21.4 | 5.8 |
| GCA | 26.2 | 0.894 | 9.2 | 3.5 |
| PEPSI | 26.8 | 0.899 | 10.2 | 3.9 |
| Ours | 27.2 | 0.901 | 10.8 | 2.5 |

**Table 5.** Performance of different inpainting methods for free-form masked images.

| Net Model | PSNR/dB | SSIM/% | Time/ms | PQ/M |
|-----------|---------|--------|---------|------|
| CE | 22.8 | 0.899 | 22.5 | 5.8 |
| GCA | 24.1 | 0.912 | 9.4 | 3.5 |
| PEPSI | 28.5 | 0.925 | 11.1 | 3.9 |
| Ours | 28.7 | 0.928 | 11.9 | 2.5 |

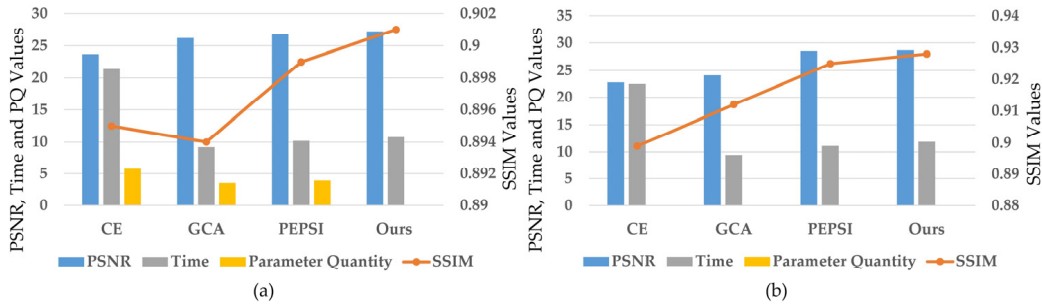

(a)  (b)

**Figure 13.** Bar charts of performance for different inpainting methods. (**a**) Metric values on square mask. (**b**) Metric values on free mask.

By utilizing a rate-adaptive convolutional layer, the proposed method can replace the standard dilated convolutional layer with fewer network parameters. Despite using significantly fewer network parameters, the proposed method achieves competitive performance with PEPSI and outperforms other conventional methods. This demonstrates that the proposed method can generate high-quality images with low hardware costs. As can be seen from the parameters, adding the Diet-PEPSI unit can significantly reduce parameters and reduce hardware requirements. In general, the model proposed in this paper is less time-consuming, requires fewer network parameters, and improves the image inpainting effect. However, we notice that the advantage of our method in SSIM is not obvious. Our method aims to calculate the correlation between missing areas and non-missing areas with the attention mechanism to inpainted images. There is no advantage in feature extraction. Moreover, the coarse-to-fine network and attention module do not learn the texture information of the bottom layer. It may lead to a poor local detail inpainting effect. The lack of supervised attention scores may lead to unreliable learning relationships. Thus, we need to introduce supervised attention to obtain reliable learning relationships in the future.

## 5. Conclusions

A GAN-based method is a useful approach in terms of image inpainting. However, there are too many parameters to train in the network; in addition, there is still room for improvement in terms of the repair result. This paper changes the usual two-stage network structure into a one-stage network, including a separate encoding network and a parallel decoding network. The decoding network is composed of a coarse path and an inpainting path. Through the parallel operation of the two paths in the decoding network, the iteration from a coarse to fine image is achieved. We reduced the size of the model parameters by introducing the Diet-PEPSI module to the encoder. We improved the transition performance at the border of the defective areas by using the truncation

distance similarity metric in CAM. We applied a region ensemble discriminator to inpaint more defective region shapes. Through comparative experiments, the algorithm in this paper can effectively reduce computational costs while achieving a satisfactory repair effect. The proposed method can be widely employed in various applications such as image generation, style transfer, and image editing. However, the algorithm still has some shortcomings. It can achieve good repair results on the datasets used in the experiments, but the image repair effect on non-training datasets is not ideal. Our future work will focus on improving the generalization of this method and further promoting it to the application of datasets in other fields.

**Author Contributions:** Conceptualization, Y.Z. and H.C.; methodology, P.Z.; software, P.Z.; validation, B.C. and X.F.; formal analysis, P.Z.; investigation, B.C. and P.Z.; resources, H.C.; data curation, B.C.; writing—original draft preparation, H.C.; writing—review and editing, Y.Z.; visualization, P.Z.; supervision, X.F.; project administration, H.C.; funding acquisition, Y.Z. All authors have read and agreed to the published version of the manuscript.

**Funding:** This work is supported by the Program for Innovative Research Team (in Science and Technology) at the University of Henan Province (No. 22IRTSTHN016), the funding scheme of Key scientific research of Henan's higher education institutions (No. 23A520010), the Key R&D and promotion Special Project of Science and Technology Department of Henan Province (No. 222102210104) and the teaching reform research and practice project of higher education in Henan Province in (No. 2021SJGLX502).

**Acknowledgments:** The authors would like to express their gratitude to the editors and anonymous reviewers for their comments and suggestions.

**Conflicts of Interest:** The authors declare no conflict of interest. The funders had no role in the design of the study; in the collection, analyses, or interpretation of data; in the writing of the manuscript; or in the decision to publish the results.

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
