# Peer review of "Image Inpainting with Parallel Decoding Structure for Future Internet"

_electronics, doi:10.3390/electronics12081872_

Round 1

Reviewer 1 Report

In this paper, the authors have proposed an improved image inpainting algorithm based on the traditional coarse-to-fine network. The background knowledge, technical detail, and experiments are well explained. To improve the quality of the paper, I have some suggestions.

1. In the abstracts, the authors should summarize the main contribution of the paper rather than explain technical detail. The same issue for Introduction, I would like to see the emphasis on the contributions.

2. The experimental results should be compared with other related works.

3. The conclusion should be rewritten to be more impressive.

Author Response

Thank you very much! We have carefully incorporated  your comments and major revision has been made on this paper.

Reviewer 2 Report

  1. The abstract should clearly state the specific problem being addressed in the paper and the proposed solution, it's too concise and not informative enough.
  2. The introduction should provide more background on the challenges and problems of Image Inpainting. Also, it should explain the importance and motivation for the proposed solution in more detail.
  3. The methods section should provide more detail on the proposed one-stage network and its components, such as the Diet-PEPSI unit and the improved contextual attention module.
  4. The methods section should provide a clear explanation for the truncation distance measure and how it improves the similarity measure in the proposed method.
  5. The experimental setup, evaluation metric and dataset should be provided more clearly.
  6. Provide more details about the comparison with other methodologies.
  7. The results and discussion section should provide more specific, quantitative evaluation of the proposed method's performance, including the reduction in number of parameters, memory, and training time, and how it improves the peak signal-to-noise ratio and structural similarity of the inpainted image.
  8. The conclusion should summarize the main contributions of the paper and its implications for the field.
  9. Check for grammar, punctuation, and citation issues. To broaden the scope of this paper, the authors should refer to some work such as: BDSS: Blockchain-based Data Sharing Scheme With Fine-grained Access Control And Permission Revocation In Medical Environment;I-Health: SDN-Based Fog Architecture for IIoT Applications in Healthcare;Enhancing Visual Coding Through Collaborative Perception
  10. Show some qualitative results, such as images, it would help to better understand the improvement made by the proposed method.

Author Response

(The authors gave the same response as above.)

Reviewer 3 Report

Introduction: Contribution of the research is missing in the introduction section. Include it.

The Related work section is presented well.

Methods: Mathematical analysis is too short, improve the content. Similarly, Include pseudocode to understand the research in a better way.

The author must include a few more dataset.

Results and discussion: The presentation of this section is good, but the comparative analysis is missing. The author did many analyses based on quantitative models. We suggest you include qualitative and comparative analysis. Compare recent year papers to show the effectiveness of the proposed model. 

Performance metrics are limited, include few more performance measures.

Include complexity analysis / statistical analysis to improve the result section.

Conclusion: What is the actual findings and how its achieved is not discussed anywhere, we recommend the author to include the finding based discussion.

References: Papers 3,5,6,7,8,21,22,25,26,27,34,36 - These references are insufficient to support this research. Include standard journal references.

Improve the figure quality.

Author Response

(The authors gave the same response as above.)

Round 2

Reviewer 2 Report

Although most comments have been well addressed, the authors should improve the paper according to the following comments, especially some references are out of date.

  1. Introduction:

a. The introduction provides a good context for the problem of image inpainting and the need for efficient algorithms. However, it could benefit from a more comprehensive literature review of recent advances in image inpainting methods and a clearer description of the limitations of existing techniques.

b. Consider stating the research question(s) or the main objective of the study more explicitly in the introduction.

  1. Methodology:

a. The Diet-PEPSI unit should be explained in more detail. Please provide a more comprehensive description of its architecture, function, and novelty in relation to existing methods.

b. The rate adaptive dilated convolution layer should be elaborated upon, explaining its purpose and advantages in the generative network.

c. The improved contextual attention module and truncation distance should be better justified. Describe how these improvements contribute to the overall performance of the proposed method compared to existing approaches.

  1. Experiments:

a. The experimental setup should be described more thoroughly. Provide more information on the datasets used, the evaluation metrics employed, and the baseline methods compared.

b. Discuss the choice of hyperparameters and the training procedure in more detail, including the number of epochs, batch size, learning rate, and any other relevant parameters.

c. Consider adding visual examples of the inpainting results, comparing the proposed method to the baseline methods. This would help illustrate the improvements in performance more effectively.

  1. Results and Discussion:

a. Present a more detailed analysis of the results, comparing the proposed method to the baseline methods in terms of various performance metrics, such as PSNR and structural similarity.

b. Discuss any potential limitations or drawbacks of the proposed method and possible ways to address them in future research.

  1. Conclusion:

a. Summarize the main findings and contributions of the study more effectively, highlighting the practical implications and significance of the proposed method in the context of image inpainting.

b. Provide suggestions for future research directions, discussing potential improvements or extensions of the proposed method.

  1. Reference: To broaden the scope of this paper, the authors should refer to some work such as A distributed covert channel of the packet ordering enhancement model based on data compression; A novel smart contract vulnerability detection method based on information graph and ensemble learning; Smart contract vulnerability detection combined with multi-objective detection
  1. Language and Presentation:

a. The paper would benefit from a thorough language review to address grammatical errors and improve clarity throughout the text.

b. Ensure that all figures and tables are properly referenced in the text, have clear captions, and are of high quality for better readability.

Author Response

Thanks very much for your valuable comments and suggestions. We have revised our manuscript accordingly.

Round 3

Reviewer 2 Report

accepted